# Discreteness Unravels the Black Hole Information Puzzle: Insights from a Quantum Gravity Toy Model

**DOI:** 10.3390/e25111479

**Published:** 2023-10-25

**Authors:** Alejandro Perez, Sami Viollet

**Affiliations:** Aix Marseille Université, Université de Toulon, CNRS, CPT, 13288 Marseille, France

**Keywords:** black hole information paradox, quantum gravity, loop quantum gravity, 98.80.Es, 04.50.Kd, 03.65.Ta

## Abstract

The black hole information puzzle can be resolved if two conditions are met. The first is that the information about what falls inside a black hole remains encoded in degrees of freedom that persist after the black hole completely evaporates. These degrees of freedom should be capable of purifying the information. The second is if these purifying degrees of freedom do not significantly contribute to the system’s energy, as the macroscopic mass of the initial black hole has been radiated away as Hawking radiation to infinity. The presence of microscopic degrees of freedom at the Planck scale provides a natural mechanism for achieving these two conditions without running into the problem of the large pair-creation probabilities of standard remnant scenarios. In the context of Hawking radiation, the first condition implies that correlations between the *in* and *out* Hawking partner particles need to be transferred to correlations between the *microscopic degrees of freedom* and the *out* partners in the radiation. This transfer occurs dynamically when the *in* partners reach the singularity inside the black hole, entering the UV regime of quantum gravity where the interaction with the microscopic degrees of freedom becomes strong. The second condition suggests that the conventional notion of the vacuum’s uniqueness in quantum field theory should fail when considering the full quantum gravity degrees of freedom. In this paper, we demonstrate both key aspects of this mechanism using a solvable toy model of a quantum black hole inspired by loop quantum gravity.

## 1. Motivation

The discussion surrounding Hawking’s information puzzle in black hole formation and evaporation [1] holds significant importance, as it serves as a testing ground for ideas aimed at establishing a coherent framework for quantum gravity. Its resolution becomes particularly crucial, as it requires the description of dynamics within the strong quantum gravitational regime near the interior singularity. This applies to both holographic-like scenarios (Here, we use this terminology to designate various approaches based on the so-called holographic principle loosely defined by Bekenstein’s idea that number of degrees of freedom inside a region are bounded by area in some way or another. This has led to various standpoints which do not necessarily agree in their details but all share a certain common ground. Examples range from works that give a fundamental status to Bousso type of entropy bounds [2], works in the context of the ADS-CFT correspondence [3], to t’Hooft’s [4] and Susskind’s ideas on complementarity [5]), where black hole entropy is a measure of the number of internal states of the black hole, and non-holographic scenarios, where the internal degrees of freedom of a black hole are not limited by the area of the horizon.

In the first case, the objective is to reproduce the Page curve [6] and elucidate the mechanism by which information escapes from the horizon while the black hole remains macroscopic. Even when the geometry near the horizon remains semiclassical in such a regime, any convincing resolution must involve the strong quantum gravity regime as, due to the no-cloning property of quantum information, it should account for the fact that no information actually reaches the near-singularity region. Perhaps with the exception of the fuzzball scenario [7], holographic approaches frequently dismiss the question of the quantum dynamics near the singularity as if it were a non-existent matter [8,9] (This was the clear position of representatives of different approaches to the problem during a panel discussion in a recent online meeting of the International Quantum Gravity Society in 2021 including N. Engelhardt, S. Giddings, A. Perez, and A., S. Raju., 2021). In the second case, a perspective in which this work is framed, it is crucial to dynamically depict the fate of the internal degrees of freedom as they evolve towards the strong Planckian regime near the singularity, since they are the carriers of the information that preserve unitarity [10,11,12].

An intriguing possibility arises in the second scenario where, following the emission of most of its mass through Hawking radiation, a black hole could potentially leave behind a Planck mass remnant. This object would contain an immense number of internal states that are correlated with the Hawking particles, thereby preserving the purity of the initial state. This proposal, known as the ‘remnant scenario’, is subject to several objections. One major concern is the existence of particle-like entities with high entropy, which would give rise to significant pair production amplitudes for these exotic objects, which would impact quantum processes at low energies (see [13] and references therein). The perceived issue stems from the assumption that remnants can be effectively described in the framework of effective field theory, a premise that becomes highly questionable when considering the extremely vast internal universes that lie beyond the horizons of old black holes from a general relativity perspective [14].

Another proposal, known as Wheeler’s bag-of-gold idea [15], is closely related, although it disregards the objection against effective field theory for the reasons mentioned earlier. Here, the remnant is presumed to harbor the purifying degrees of freedom within a vast internal ‘universe’. These degrees of freedom would gradually escape into the nearly flat external spacetime as extremely low-energy particles after the black hole has completed its evaporation. However, there are significant challenges associated with this proposal. If a black hole begins with a macroscopic mass *M* and undergoes evaporation through standard Hawking radiation, following semiclassical expectations until its mass nears the Planck mass $m_{p}$, energy conservation dictates that the individual components of the purifying radiation emitted by the remnant possess energies smaller than $m_{p}^{3}/M^{2}$ for a remnant of Planckian dimensions. At the superficial level, it is already difficult to conceive how such long-wavelength particles could be emitted from an object that appears Planckian from an external perspective. At a deeper level, and despite some recent interest [16] in the context of loop quantum gravity, simple entropic considerations make such a proposal highly improbable within a theory that incorporates fundamental discreteness. Specifically, as mentioned before (and illustrated in Figure 1), the Hawking partners that were initially correlated with the Hawking radiation are forced to hit the strong Planckian regime near the singularity residing within and thus interact with a new enormous number microscopic degrees of freedom: the atomistic structure of spacetime in such a theory. From an entropic standpoint, this would necessitate an exceedingly delicate conspiracy for these excitations to reorganize into extraordinarily long-wavelength particles within quantum field theory as they emerge from the remnant (for further discussion, see [10]).

The literature explores several additional possibilities worth considering. One such possibility is that the internal universe of the remnant remains causally disconnected from the external universe where black hole evaporation occurs, without any leakage of particles. This scenario would effectively break unitarity for the outside world. Another possibility is that when gravity is incorporated, quantum mechanics may require modifications wherein unitarity ceases to be a fundamental law (as discussed from Surdarsky’s perspective in [10] and related references). Furthermore, it is conceivable that certain aspects of the semiclassical language employed to describe the problem are inadequate, and the final state of evaporation may preclude a meaningful mean field spacetime representation (as highlighted in [17,18] and more recently in [19,20,21,22]). In such cases, the non-local nature of quantum mechanics and gravity could blur the distinction between the inside and outside regions [23]. Presently, there are no substantial objections to these possibilities. To ensure clarity in presenting our specific proposal, we have chosen to relegate the discussion of these aspects to a secondary role, with the hope that readers can obtain a more comprehensive understanding from the contributions of other authors in this Special Issue. Perhaps, in the future, a consensus may emerge from the diverse perspectives that currently appear to conflict with one another.

Our perspective offers a straightforward and intuitive resolution of the paradox that aligns with other well-established descriptions of standard physical systems (like standard matter in statistical mechanics), in which information is degraded, yet not lost, due to the presence of microscopic granularity [24]. The idea has been recently shown to be viable in certain models of analogue gravity [25]. The proposal is in sharp contrast with the standard remnant scenario, as the Planck-scale microscopic degrees of freedom purifying the Hawking radiation are not exclusively localized within a particle-like object but instead are distributed throughout the fabric of spacetime after the complete evaporation of the black hole. In this way, our proposal avoids the objections raised against standard remnants. This possibility is based on the assumption that there is a quantum evolution across the singularity (in the context of loop quantum gravity, the paradigm was first put forward in [26] and includes more recent black hole to white hole transition models [27]). The physical situation representing black hole formation and evaporation, of what we call the Ashtekar–Bojowald paradigm, is illustrated by the Penrose diagram shown in Figure 2 where the singularity is replaced by a region where no spacetime description is possible and a full quantum gravity treatment becomes mandatory.

A pictorial representation of the scenario for purification that we propose is given in Figure 1. Entanglement between the Hawking radiation going to infinity at ${ℐ}^{+}$ and inside partners maintain unitarity in the semiclassical quantum field theory description that is expected to be valid while the black hole is macroscopic: in our illustration, we assume this is so around the instant defined by the Cauchy surface $\Sigma_{1}$. Inside partners (wavy lines on the left of the horizon) fall into the quantum gravity region (what, classically, we call the singularity) and transfer their entanglement to defect-like Planckian degrees of freedom that cannot be described in an effective field theory framework (they are discrete excitations in an fundamentally discrete theory with no field theoretic analog). These are represented by dotted lines in the quantum gravity region. The evolution between $\Sigma_{1}$ and $\Sigma_{3}$ in the full quantum gravity theory includes the discrete fundamental degrees of freedom and is thus unitary. In contrast with fuzzballs, the amount of information that can be coded in these defects grows with the number of defects involved—which can be arbitrary depending on the internal extension of the black hole, e.g., one per Planck volume—and hence is by no means bounded by the area of the corresponding horizon. Information would seem to be lost in any effective description that neglects such microscopic defects. Now, for the scenario to be consistent with energy conservation, the Bondi mass of the system for retarded times after $u_{0}$ (retarded time of complete semiclassical evaporation) must be close to zero or about the Planck mass. This is possible if the macroscopic emergent geometry is degenerate, i.e., it corresponds to many microscopically inequivalent configurations. This is a feature expected to be present in certain approaches to quantum gravity. In the case of loop quantum gravity, such property is precisely the one that explains the origin of black hole entropy in the semiclassical regime [28,29,30,31] (for reviews see [11,32] for more discussion of this specific feature, see [10]) (The existence of such fundamental granularity is also a potential source of interesting phenomenology: as argued in [33] diffusion effects could provide a natural paradigm for the generation of dark energy [34,35], and (under suitable circumstances and with natural assumptions) provide the seeds for the inhomogeneities observed at the CMB in cosmology [33] or explain the origin of dark matter [33,36]).

A fundamental description of the previous mechanism remains out of reach at the present stage of development of approaches to quantum gravity with Planckian granularity. However, it is possible to make the previous scenario concrete in simplified models. The interior region r<2M of a Schwarzschild black hole of mass *M* can be seen as a homogeneous anisotropic cosmological model where the *r* = constant surfaces (in the usual Schwarzschild coordinates) are Cauchy surfaces of homogeneity: any two arbitrary points can be connected along orbits of the isometry group that involves spacelike translations along the staticity Killing field $\xi =\partial_{t}$ and the rotations associated with spherical symmetry. Models with these isometries will be referred to as Kantowski–Sachs (KS) models [37]. They include not only the Schwarzschild black hole interior geometry (vacuum case) but also the Reissner–Nordstrom black hole interior geometry (in the Einstein–Maxwell case) and other solutions depending of the type of matter that one decides to couple to the system. In a recent paper, we showed that such Kantowski–Sachs models (with a massless scalar field coupling) define a natural toy model capturing aspects of the dynamics and back-reaction of matter near the singularity of realistic black holes that Hawking radiate and evaporate [38]. The model is quantized *á la loop* in that paper, and it is shown that certain microscopic degrees of freedom (not relevant for coarse observers) arise as a consequence of peculiar representation of the algebra of observables used. Such, at first appearance, exotic representation is indeed the natural one when it comes to the construction of the background independent quantization of gravity in loop quantum gravity [39]. The presence of microscopic degrees of freedom that remain hidden at low energies is an expected generic feature in the full quantum gravity regime (Even when the continuum limit remains an open issue in the framework of loop quantum gravity, the are various potential sources of micrsocopic degeneracy in the structure of the theory. For instance, consistent regularizations of the Hamiltonian constraint all seem to share the property of being insensitive to certain microscopic details of the quantum states that could code informations. These defect-like structures are related to the fundamental degeneracy of the quantum geometry at the microscopic level. There are studies showing that in the emergent low-energy regime, some degrees of freedom would decouple as they would appear highly massive [40,41,42].).

The existence of such a hidden sector (which represents a sort of quantum hair) is central in the scenario for purification of the Hawking radiation in black hole evaporation proposed in [24]. In the present paper, we use the simple black hole model recalled above to illustrate the key ingredients of the general scenario: firstly, we review how microscopic degrees of freedom arise in the quantization, secondly, we show how quantum correlations between matter excitations and such microscopic degrees of freedom develop unavoidably during dynamical evolution. This suggests that the microscopic physics does play an important role in discussions about the fate of unitarity in black hole formation and evaporation, and that any effective field theory approach to the unitarity question that avoids confronting the details of the dynamics near the singularity misses a central aspect and thus cannot provide a complete picture of the resolution of the puzzle.

The article is organized as follows. In Section 2, we review the definition of the simplified model for the dynamics of matter and geometry inside of a spherically symmetric black hole. In this section, we basically reproduce the results and aspects of the presentation of [38]. We give the definition of the mini-superspace model of interest and review its polymer quantization. In Section 3, we use the simplified setup to construct a minimalistic model of the dynamics of a Hawking pair on our simplified black hole setup. The back reaction of the in-falling partner is suitably taken into account, as it approaches the strong field regime near the singularity where the coupling between geometry and matter becomes most important. The outside partner dynamics are assumed to be trivial (idealizing the weak back reaction for wave packets evolving toward infinity; completely neglected in the framework of the Hawking effect). However, the model suitably keeps track of the entanglement with the inside: the full dynamics are unitary. In Section 4, we present the key results supporting our claims. We end with a discussion of the paper in Section 5.

## 2. The Model

We start by reviewing the model presented in [38] that we will use later on in this work to calculate the back reaction of a Hawking pair as one of the partners falls into the black hole singularity. In a description of an *r* equal constant slicing of the interior of a Schwarzschild black hole, a free test scalar excitation (with no angular momentum) is well approximated as it approaches the singularity through a translational invariant wave function, as the expansion in the spacelike Killing direction $\xi =\partial_{t}$ diverges for r→0. This means that zero angular momentum test particles can be approximated by the type of excitations that can be accommodated in the dynamical framework of the KS cosmologies (at least in the sense of a near singularity approximation) (One can be quantitative about this intuition as follows: free test particles with four wave vector $k^{a}$ on the Schwarzschild background are associated with the conserved Killing energy $E\equiv -k^{a}\xi_{a}$. We are assuming that the particle has zero angular momentum, which implies that its wave function is already translational invariant in the directions transversal to $\xi^{a}$ on the *r*-slices. The wave function can only vary in the direction of the Killing $\xi^{a}$ and the component of the physical momentum in this direction is given by
(1)$$p_{\xi }\equiv \frac{k^{a}\xi_{a}}{\sqrt{\xi \;\cdot \;\xi }}=-E\sqrt{\frac{r}{2M-r}},$$
which vanishes in the limit r→0. The wave length of such a particle diverges, and thus particles without angular momentum are increasingly well-represented by translational invariant excitations as one approaches the singularity). This simple implication, deduced from the idealized notion of test particle, can be made more precise by looking at the analogous features of scalar field excitations (solutions of the Klein–Gordon equation). Indeed, the simplistic argument given here can be made precise, as shown in [38]. Therefore, in what follows, we will review the construction of a quantum model of the black hole interior that can accommodate such scalar excitations and their back reaction. Such model will set the stage where an idealized Hawking pair will be defined and the quantum information dynamical details relevant for our discussion will be computed.

### 2.1. Symmetry Reduced Covariant Phase Space

Here, we briefly report the construction of the model introduced in [38] (refer to that paper for more details). The first step towards canonical quantization is to define the Hamiltonian description of the model. It is well known that for a spherically symmetric and static spacetime, the line element can be written without any loss of generality as
(2)$$ds^{2}=-f\left(r\right)dt^{2}+h\left(r\right)dr^{2}+r^{2}d\Omega^{2}\;.$$
It follows that the Einstein–Hilbert action (with the appropriate boundary term that renders it differentiable: the usual Gibbons–Hawking–York boundary term) becomes
(3)$$S_{\mathrm{geo}}=\frac{1}{16\pi \ell_{p}^{2}}\left[\int_{R}d^{4}x\sqrt{-g}R+2\int_{\partial R}K\right]=\frac{\ell_{0}}{2\ell_{p}^{2}}\int dr\left(\sqrt{fh}+\sqrt{\frac{f}{h}}+\frac{\dot{f}r}{\sqrt{fh}}\right),$$
where $\ell_{p}^{2}=G$ (in ℏ=1 units) and $\ell_{0}$ is a infra-red cut off in the Killing time *t* necessary to make the action of the symmetry reduced model finite (for more details, see [38]). From which we read the Lagrangian ${ℒ}_{\mathrm{geo}}$ of the spacetime subsystem
(4)$${ℒ}_{\mathrm{geo}}=\frac{\ell_{0}}{2\ell_{p}^{2}}\left(\sqrt{fh}+\sqrt{\frac{f}{h}}+\frac{r\dot{f}}{\sqrt{fh}}\right).$$
On the other hand, we will couple the system to a massless scalar field by adding the matter action
(5)$$S_{m}=-\frac{1}{2}\int_{R}d^{4}x\sqrt{-g}\partial_{a}\phi \partial^{a}\phi =-2\pi \ell_{0}\int drr^{2}{\dot{\phi }}^{2}\sqrt{\frac{f}{h}}.$$
The conjugate momenta to f,h and ϕ are given by
(6)$$p_{f}=\frac{\ell_{0}}{2\ell_{p}^{2}}\frac{r}{\sqrt{fh}}\;,\;p_{h}=0\;\mathrm{and}\;p_{\phi }=-4\pi r^{2}\ell_{0}\sqrt{\frac{f}{h}}\dot{\phi },$$
and the primary Hamiltonian, defined by $H=\dot{f}p_{f}+\dot{h}p_{h}+\dot{\phi }p_{\phi }-{ℒ}_{\phi }-{ℒ}_{\mathrm{geo}}$, becomes
(7)$$H_{1}=-\frac{\ell_{0}}{2\ell_{p}^{2}}\frac{f(h+1)}{\sqrt{fh}}-\frac{hp_{\phi }^{2}}{8\pi r^{2}\ell_{0}\sqrt{fh}}\;.$$
From the expression of the conjugate momenta (Equation 6), we identify the constraints
(8)$$\xi \equiv p_{f}-\frac{\ell_{0}}{2\ell_{p}^{2}}\frac{r}{\sqrt{fh}}=0\;\mathrm{and}\;p_{h}=0,$$
and the secondary Hamiltonian
(9)$$H_{2}=H_{1}+\lambda \xi +\eta p_{h}\;,$$
where λ and η are Lagrange multipliers. One can show that the stability of the two constraints (Equation 8) can be ensured by fixing the associated Lagrange multipliers, i.e., the constraints (Equation 8) are second class and can be explicitly solved, leading to
(10)$$p_{h}=0\;\mathrm{and}\;h=\frac{\ell_{0}^{2}}{4\ell_{p}^{4}}\frac{r^{2}}{fp_{f}^{2}}.$$
Thus, the secondary Hamiltonian (Equation 9) reduces to
(11)$$H_{2}=-\frac{1}{r}\left(fp_{f}+\frac{1}{16\pi \ell_{p}^{2}}\frac{p_{\phi }^{2}}{fp_{f}}+\frac{\ell_{0}^{2}}{4\ell_{p}^{4}}\frac{r^{2}}{p_{f}}\right)\;.$$
The previous Hamiltonian encodes the KS dynamics of geometry coupled to a massless scalar field. The relevant solutions for physical applications correspond to small departures from the vacuum Schwarzschild solutions representing macroscopic black holes with scalar field perturbation falling inside. The system is simplified further by focusing on what we call the deep interior region r≪M where *M* is the mass scale defined by the corresponding black hole solution perturbed by the presence of matter. It is in this regime that the solutions of the KS system faithfully describe the dynamics of a spherically symmetric scalar perturbation (representing, for instance, a Hawking particle) as it falls towards the interior singularity. It is in this regime that the KS Hamiltonian evolution given by (Equation 11) matches the test-field evolution (the Klein–Gordon solutions on the Schwarzschild background fixed non dynamical background) and incorporates, as a simplified model, aspects of the back-reaction that are expected to become more important as one approaches the singularity. As shown in [38], the simplification simply amounts to dropping the last term in the previous Hamiltonian, namely
(12)$$H_{\mathrm{di}}=-\frac{1}{r}\left(fp_{f}+\frac{1}{16\pi \ell_{p}^{2}}\frac{p_{\phi }^{2}}{fp_{f}}\right).$$
This toy theory reflects the dynamics of the leading order in an expansion near r=0. In order to recover the structure of a gauge theory, with a clear analogy with the full theory of LQG, it is convenient to ‘reparametrize’ the system by promoting the area radius *r* to a degree of freedom with conjugate momentum $p_{r}$ and add a scalar constraint $C=p_{r}-H_{2}=0$. Introducing the *deep interior variables*
(13)$$m=-fp_{f}\;\;\;\mathrm{and}\;\;\;p_{m}=-\log\left(-f\right)\;,$$
and adopting the area *a* of the surfaces of constant *r* as time
(14)$$a=4\pi r^{2}\;\mathrm{and}\;p_{a}=\frac{p_{r}}{8\pi r}\;,$$
one has
$$\begin{array}{cc} \left\{m,p_{m}\right\} & =1\;,\;\{a,p_{a}\}=1\;, \end{array}$$
and
(15)$$\{\phi ,p_{\phi }\}=1,$$
with all the other Poisson brackets equal to zero. The Hamiltonian constraint describing the deep interior regime is
(16)$$\begin{array}{c} C_{a}=p_{a}+\frac{1}{2a}\left(m+\frac{p_{\phi }^{2}}{16\pi \ell_{p}^{2}m}\right)\approx 0. \end{array}$$
The previous constraint defines the classical dynamical equation of the model. In the quantum theory—and in the loop representation that mimics the one used in the full LQG framework—the area variable will evolve in discrete steps. A side effect of such discreteness is the appearance of discrete quantum hair: black hole states are labeled by macroscopic quantum numbers corresponding to the eigenvalues of $p_{\phi }$ and *m*, but also by a microscopic quantum number ϵ (quantum hair).

### 2.2. Sketch of the Schroedinger Quantization

For comparison, let us start by reviewing the Schroedinger quantization of the system presented in the last section. In the standard Schroedinger representation, one would quantize the phase space of Section 2.1 by promoting the variables $a,m,p_{a},p_{m}$ to self adjoint operators
$$\begin{array}{c} \;\;\;\;\;\;\;\;\begin{array}{ccc} & & \hat{m}\;\psi \left(m,p_{\phi },a\right)=m\psi \left(m,p_{\phi },a\right),\\ & & {\hat{p}}_{m}\psi \left(m,p_{\phi },a\right)=-i\partial_{m}\psi \left(m,p_{\phi },a\right), \end{array}\;\begin{array}{ccc} & & \hat{a}\;\psi \left(m,p_{\phi },a\right)=a\psi \left(m,p_{\phi },a\right),\\ & & {\hat{p}}_{a}\psi \left(m,p_{\phi },a\right)=-i\partial_{a}\psi \left(m,p_{\phi },a\right), \end{array}\;\begin{array}{ccc} & & {\hat{p}}_{\phi }\psi \left(m,p_{\phi },a\right)=p_{\phi }\psi \left(m,p_{\phi },a\right),\\ & & \hat{\phi }\;\psi \left(m,p_{\phi },a\right)=i\partial_{p_{\phi }}\psi \left(m,p_{\phi },a\right), \end{array} \end{array}$$
in the kinematical Hilbert space is ${ℋ}_{S}={ℒ}^{2}\left(R^{3}\right)$, equipped with the usual inner product
(17)$$\langle \psi_{1},\psi_{2}\rangle =\int_{-\infty }^{+\infty }\int_{-\infty }^{+\infty }\int_{-\infty }^{+\infty }\bar{\psi_{1}\left(m,p_{\phi },a\right)}\psi_{2}\left(m,p_{\phi },a\right)dmdp_{\phi }da,$$
where we have chosen the momentum representation for the scalar field for convenience (as $p_{\phi }$ is one of the constants of motion of the system). Eigenstates of the $\hat{a}$ operator are interpreted as distributions (they are not in the Hilbert space) and one usually writes
(18)$$\hat{a}|a\rangle =a|a\rangle$$
with a∈R and forms an orthonormal basis
(19)$$\langle a,a^{\prime }\rangle =\delta (a,a^{\prime }).$$
The dynamics are imposed by solving the Hamiltonian constraint (Equation 16) which, in the present representation, takes the precise form of a Schroedinger equation in the area variable *a*, namely
(20)$$\left[-i\hbar \frac{\partial }{\partial a}+\frac{1}{2a}\left(m+\frac{p_{\phi }^{2}}{16\pi \ell_{p}^{2}m}\right)\right]\psi \left(m,p_{\phi },a\right)=0.$$
As usual, solutions of the constraint are certainly not square integrable in the *a*-direction; thus, physical states are outside of the kinematical Hilbert. The physical Hilbert space is defined as the space of square integrable functions of *m* and $p_{\phi }$ at fixed time *a*—${ℋ}_{\mathrm{phys}}={ℒ}^{2}\left(R^{2}\right)$—with inner product
(21)$${\langle \psi_{1}\left(a\right),\psi_{2}\left(a\right)\rangle }_{\mathrm{phys}}=\int_{-\infty }^{+\infty }\int_{-\infty }^{+\infty }\bar{\psi_{1}\left(m,p_{\phi },a\right)}\psi_{2}\left(m,p_{\phi },a\right)dmdp_{\phi },$$
which is preserved, i.e., it is independent of *a*, by the Schroedinger equation (evolution is unitary in *a*).

Two important remarks are in order: First, note that we are formulating in detail the dynamics of the system in the near singularity approximation. The physical reason for this is that (as argued previously) it is only in this approximation that the system can be compared with a (spherically symmetric) black hole with spherically symmetric excitations falling inside. A side gain is also the simplification of the dynamics which will allow for a simpler quantization and the analysis of the possibility of well-defined dynamics across the singularity when we undergo the LQG inspired quantization. One could, however, consider the quantization of the minisuperspace system without the near-singularity approximation. In that case one would need to write a Schroedinger equation using the Hamiltonian (Equation 11), now genuinely time-dependent (*r*-dependent), for which unitary evolution would involve path-ordered exponentials (as the Hamiltonian does not commute with itself at different *r* values). In addition, one would need to work with either $r,f,p_{r},p_{f}$ variables or $a,f,p_{a},p_{f}$ variables without the luxury of the simplifications introduced by the use of the near-singularity variables (Equation 13).

### 2.3. The Loopy Quantization

We now define a representation of the phase space variables that incorporates a key feature of the full theory of LQG: the area quantization. This representation closely mimics the structure of the quantum theory in the fundamental theory in such a way that the area variable *a* acquires a discrete spectrum. Mathematically, this is achieved by replacing the ${ℒ}^{2}$ structure of the inner product in the variable $p_{a}$ with the inner product of the Bohr compactification of the $p_{a}$ phase space dimension. This is analogous to what is carried out in the full theory of loop quantum gravity that uses the Ashtekar–Barbero connection-variables [43,44] as the starting point. There, the connection is not represented as a fundamental operator but only its holonomy (an exponentiated version of the connection in essence) in the definition of the Hilbert space [39]. More precisely, one replaces the kinematical inner product in the Schroedinger representation (Equation 17) as follows
(22)$$\langle \psi_{1},\psi_{2}\rangle =\lim_{\Delta \to +\infty }\frac{1}{2\Delta }\int_{-\Delta }^{+\Delta }\left(\int_{-\infty }^{+\infty }\bar{\psi_{1}\left(m,p_{\phi },p_{a}\right)}\psi_{2}\left(m,p_{\phi },p_{a}\right)dmdp_{\phi }\right)dp_{a}\;.$$
With this inner product, periodic functions of $p_{a}$ with an arbitrary period are normalizable and the conjugate *a*-representation acquires the property of discreteness in a way that closely mimics the structure of the fundamental theory of loop quantum gravity [45]. In particular, eigenstates of $\hat{a}$ exist
(23)$$\hat{a}|a\rangle =a|a\rangle$$
with a∈R. These states form an orthonormal basis with inner product
(24)$$\langle a,a^{\prime }\rangle =\delta_{a,a^{\prime }},$$
in contrast with (Equation 19). Discreteness of the spectrum of $\hat{a}$ comes at the price of changing the kinematical Hilbert space structure in a way that prevents the infinitesimal translation operator ${\hat{p}}_{a}$ from existing (this is because the inner product (Equation 22) is not weakly continuous under the action of translations: according to (Equation 24) an infinitesimally translated state is orthogonal to the original state). Instead, only finite translations (quasi periodic functions of $p_{a}$) can be represented as unitary operators in the polymer Hilbert space. Their action on the *a*-basis is given by
(25)$$\hat{e^{i\lambda p_{a}}}\psi \left(m,p_{\phi },a\right)=\psi \left(m,p_{\phi },a+\lambda \ell_{p}\right).$$
Eigenstates of the finite translations (or shift operators) exist and are given by wave functions supported on discrete *a*-lattices. Namely,
(26)$$\psi_{k,\epsilon }\left(a\right)\equiv \left\{\begin{array}{ccc} & \exp(ika) & \;\;\;\mathrm{if}\;\;\;a\in \Gamma_{\epsilon ,\lambda }\equiv {\left\{\left(\epsilon +n\lambda \right)\ell_{p}^{2}\in R\right\}}_{n\in Z}\\ & 0 & \;\;\;\mathrm{otherwise} \end{array}\right.$$
where the parameter ϵ∈[0,λ)∈R. The discrete lattices denoted $\Gamma_{\epsilon ,\lambda }$ are the analog of the spin-network graphs in LQG with the values of *a* on lattice sites the analog of the corresponding spin labels. With all this, one has (using (Equation 25)) that
(27)$$\hat{e^{i\lambda p_{a}}}\psi_{k,\epsilon }\left(a\right)=e^{i\lambda k}\psi_{k,\epsilon }\left(a\right).$$
Note that, unlike the Schroedinger representation where the eigen-space of the momentum operator is one dimensional, the eigen-spaces of the translation operator (labeled by the eigenvalue $e^{i\lambda k}$) are infinite dimensional and non separable. This is explicit from the independence of the eigenvalues of the continuous parameter ϵ∈[0,λ) labeling eigenstates. Such huge added degeneracy in the spectrum of the shift operators is a general feature of the polymer representation. We will show that this degeneracy can show up in Dirac observables of central physical importance, such as the mass operator in Section 2.5.

### 2.4. Quantum Dynamics

In the full theory, a new perspective on the regularization issue has been introduced, which is motivated by the novel mathematical notion of generalized gauge covariant Lie derivatives [46] and their geometric interpretation allowing for the introduction of a natural regularization (and subsequent) anomaly-free quantization of the Hamiltonian constraint [47]. Even when the procedure does not eliminate all ambiguities of quantization (choices are available in the part of the quantum constraint responsible for propagation [48]), the new technique drastically reduces some of them in the part of the Hamiltonian that is more stringently constrained by the quantum algebra of surface deformations.

The primary focus of our discussion is to highlight the similarity in the impact of the analogous process when applied to our symmetry-reduced Hamiltonian. Our classical Hamiltonian constraint features linearity in the variable $p_{a}$, whose corresponding Hamiltonian vector field has a clear geometric interpretation of infinitesimal translations in *a* which in the quantum theory can only be represented by finite translations or shifts. Thus, one makes the replacement
(28)$$\lambda {\hat{p}}_{a}\;\;\;⟶\;\;\;\hat{e^{i\lambda p_{a}}}.$$
Now, in order to maintain geometric compatibility with the Schroedinger equation, one exponentiates the second term in the classical Hamiltonian (Equation 16), obtaining the well-known unitary evolution operator producing finite area evolution. The quantum constraint preserving the geometric consistency with the Schroedinger equation is
(29)$$\underset{\begin{array}{c} \mathrm{finite}\;\mathrm{areatime}\\ \mathrm{translation} \end{array}}{\underset{︸}{\exp(i\lambda p_{a})}}|\psi \rangle -\overset{\begin{array}{c} \mathrm{finite}\;\mathrm{areatime}\\ \mathrm{unitary}\;\mathrm{evolution}\;\mathrm{operator} \end{array}}{\overset{︷}{\exp\left(\frac{i}{2}\log\left(\frac{a+\lambda \ell_{p}^{2}}{a}\right)\left(m+\frac{p_{\phi }^{2}}{16\pi \ell_{p}^{2}m}\right)\right)}}\;|\psi \rangle =0,$$
whose action is well defined in the polymer representation and whose solutions are easily found (by acting on the left with $\langle m,p_{\phi },a|$) to be wave functions satisfying the discrete dynamics given by
(30)$$\psi \left(m,p_{\phi },a+\lambda \ell_{p}^{2}\right)=e^{\frac{i}{2}\log\left(\frac{a+\lambda \ell_{p}^{2}}{a}\right)\left(m+\frac{p_{\phi }^{2}}{16\pi \ell_{p}^{2}m}\right)}\psi \left(m,p_{\phi },a\right).$$
The physical Hilbert space is defined via the usual inner product at fixed (discrete) time *a* via
(31)$${\langle \psi_{1}\left(a\right),\psi_{2}\left(a\right)\rangle }_{\mathrm{phys}}=\int_{-\infty }^{+\infty }\int_{-\infty }^{+\infty }\bar{\psi_{1}\left(m,p_{\phi },a\right)}\psi_{2}\left(m,p_{\phi },a\right)dmdp_{\phi },$$
which is independent of the lattice sites as required (a property identified with the unitarity of the dynamics generated by the quantum constraint). More precisely the physical inner product is a constant of the quantum motion associated with the full history represented by the lattice $\Gamma_{\epsilon ,\lambda }$ as implied by unitarity. Explicitly, one has
(32)$${\langle \psi_{1}\left(a\right),\psi_{2}\left(a\right)\rangle }_{\mathrm{phys}}={\langle \psi_{1}\left(a+\lambda \right),\psi_{2}\left(a+\lambda \right)\rangle }_{\mathrm{phys}}\;\;\;\forall \;\;\;a\in \Gamma_{\epsilon ,\lambda }.$$
Ambiguities of regularization that are usually associated with the polymerization procedure are thus completely absent in this model. The reason is the linear dependence of the Hamiltonian constraint in the polymerized variable, which allows for a regularization fixed by the geometric interpretation of the classical Hamiltonian vector field associated with the corresponding variable. However, ambiguities remain when one studies the evolution across the *would-be-singularity* of the Kantowski–Sachs model at a=0 [38].

Now, we would like to concentrate on the evolution when we are away from the a=0. In such a regime, the one-step evolution (Equation 30) can be composed to produce the arbitrary initial to final area evolution
(33)$$\psi \left(m,p_{\phi },\epsilon +n\lambda \right)={\left(\frac{\epsilon +n\lambda }{\epsilon +q\lambda }\right)}^{\frac{im}{2}\left(1+\frac{p_{\phi }^{2}}{16\pi \ell_{p}^{2}m^{2}}\right)}\psi \left(m,p_{\phi },\epsilon +q\lambda \right),$$
for arbitrary integers n,q>1.

The polymer dynamics that arise from the geometric action of the quantum constraint (Equation 29) enjoys the appealing feature of being closely related to the dynamics that one would obtain in the continuum Schroedinger representation. This statement can be made precise as follows: any solution to the Schroedinger Equation (Equation 20) induces, on any given lattice $\Gamma_{\epsilon ,\lambda }$ a solution of (Equation 29). Conversely, physical states of the polymer theory represent a discrete sampling of the continuum solutions of (Equation 20). However, the Schroedinger evolution is ill defined at the singularity a=0 due to the divergence of the 1/a factor in front of the second term of (Equation 20). The polymer representation allows for a well-defined evolution across the singularity thanks to the deviations from the 1/a behaviour introduced by the analog of the ‘inverse-volume’ corrections [45]. As discussed in [49], these corrections are ambiguous (a fact that should not be surprising, given the expectation that the classical theory cannot guide us all the way to the deep UV in QFT). Instead of proposing one particular UV extension, as in the example shown in [38], where Thiemann regularization was used, one might simply keep all possibilities open and assume that the corresponding operator is regularized in the relevant region by some arbitrary function log(a)→τ(a). In regions where τ(a)=log(a), the quantum evolution leads to semiclassical equations that exactly match Einstein’s equations in the KS sector, when semiclassical states are considered, and the Schroedinger equation away from a=0.

### 2.5. Quantum Hair: The Mass Operator (in the Vacuum Case)

As mentioned at the end of Section 2.3, the huge degeneracy introduced by the presence of the ϵ sectors plays an important role in the spectrum of physical observables. In this section, we will study this degeneracy for the mass operator in the vacuum case. The mass of the black hole (in the vacuum case) is given by the following expression in terms of the variables (Equation 13) (see [38] for details)
(34)$$M=\alpha \left(a\right)m^{2}e^{p_{m}}.$$
where $\alpha \left(a\right)\equiv 2\sqrt{4\pi }\ell_{p}^{4}/\left(\ell_{0}^{2}\sqrt{a}\right)$. For the ordering $m\exp p_{m}m$ the eigenstates are
(35)$$|M\rangle =\sum_{a\in \Gamma_{\epsilon ,\lambda }}\int \phi_{M}\left(p_{m},a\right)|p_{m}\rangle |a\rangle dp_{m},$$
where the sum runs over the discrete lattice $\Gamma_{\epsilon ,\lambda }$ defined in (Equation 26) when introducing the dynamical constraint (Equation 29), with
(36)$$\phi_{M}\left(p_{m},a\right)\equiv \langle p_{m},a|M\rangle =\sqrt{\frac{2\sqrt{M}}{\alpha (a)}}e^{-\frac{p_{m}}{2}}J_{1}\left(2\sqrt{\frac{M}{\alpha (a)}}e^{-\frac{p_{m}}{2}}\right),$$
where $J_{1}$ is a Bessel function. One can explicitly verify that the quantum dynamics (Equation 29) preserve the eigenstates. The spectrum of the mass operator is continuous. It was argued in the context of the full LQG theory in [10,12,24] that the eigenspaces of the mass should be infinitely degenerate due to the underlying discrete structure of the fundamental theory and the existence of defects that would not be registered in the ADM mass operator. Interestingly, the conjectured property is illustrated explicitly in our simple toy model as the eigenvectors (Equation 35) for a given eigenvalue *M* there are infinitely many and labeled by a continuum parameter. More precisely, they are associated with wave functions of the form (Equation 36) supported on lattices with different values of ϵ. Thus eigenstates of the mass should then be denoted |M,ϵ〉 with orthogonality relation
(37)$${\langle M,\epsilon |M^{\prime },\epsilon^{\prime }\rangle }_{\mathrm{phys}}=\delta \left(M,M^{\prime }\right)\delta_{\epsilon ,\epsilon^{\prime }},$$
where $\delta_{\epsilon ,\epsilon^{\prime }}$ is the Kronecker delta symbol. The existence of such a large degeneracy is a generic feature of the polymer (or loopy) representation. Even when this is a toy model of quantum gravity, this feature is likely to reflect a basic property of the representation of the algebra of observables in the full LQG context. Here, we are showing that the mass operator is hugely degenerate, suggesting that the usual assumption of the uniqueness of the vacuum in background dependent treatments of quantum field theory might fail in the full quantum gravity context. The present conclusions are independent of factor ordering ambiguities.

## 3. Modelization of a Hawking Pair on Our Quantum Geometry

Outgoing Hawking excitations detected in normalizable wave packets at future null infinity are expected to evolve unitarily—independently of the inside degrees of freedom—when they become sufficiently separated from the black hole. These excitations are, however, entangled with the inside falling partner. The back reaction of the inside partner on the quantum geometry is expected to be strong as the particle falls toward the singularity. Instead, the back reaction of the outside partner is expected to be weak, as the evolution happens in the weak gravitational field region (the back reaction is completely neglected in the original Hawking calculation for these reasons). The setup we introduce in this section incorporates a simplified version for the back reaction for the inside partner (based on the model of the previous section). The dynamical evolution for the outside partner is simplified by assuming it to be trivial: the outside Hamiltonian vanishes. The key feature is, however, the entanglement in the initial state and the transfer of information to the microscopic structure of the quantum geometry, which will be shown in Section 4.

We have argued in the introduction about why such a mechanism involves primarily the inside quantum dynamics where the strong field regime activates the interaction with the Planckian discreteness. This argument is valid in general (independent of our simplifying assumptions) and realized in particular in our simple model. Accordingly, in this section, we use the model introduced in the previous section to describe the non-trivial dynamics of the inside partner evolving towards the strong quantum gravity region near the singularity, while letting the outside partner evolve freely with a trivial Hamiltonian. The exact dynamics of the corresponding density matrix of the full system as well as the one corresponding to the relevant reduced density matrices of suitable subsystems will be written explicitly.

### 3.1. Construction of the Initial State

In order to simplify the description of the matter part as much as we can, we will consider that the *inside* matter can be in two different orthogonal states |+〉 or $|-\rangle \in {ℋ}_{in}$ (with subscript in standing for the *inside* Hilbert space of the scalar degrees of freedom) which are eigenvectors of ${\hat{p}}_{\phi }$ of eigenvalue $p_{+}$ and $p_{-}$, respectively. On the other hand, we will consider that the outgoing part of the matter can be in two different orthogonal states |+〉 or $|-\rangle \in {ℋ}_{out}$ (with subscript out standing for the *outside* Hilbert space of the scalar degrees of freedom). As the outside outgoing partner evolves in regions of low curvature, where the type of effects we are focusing on should be very small, we model such evolution by a trivial Hamiltonian. This simple description of the matter is sufficient to write down a state for the total matter where the ingoing particle and the outgoing one are initially maximally entangled, and to study how the dynamic modifies these correlations. We can now write down the total Hilbert space describing the geometrical and the matter degrees of freedom. It can be written as
(38)$${ℋ}_{total}=\underset{\mathrm{geometry}}{\underset{︸}{{ℋ}_{m}⊗{ℋ}_{\epsilon }}}⊗\overset{\mathrm{matter}}{\overset{︷}{{ℋ}_{in}⊗{ℋ}_{out}}}\;,$$
where we have organized the Hilbert space of geometry as the tensor product of normalizable functions ${ℋ}_{m}$ of the configuration variable *m*, defined in (Equation 13), on a given lattice times the separable truncation ${ℋ}_{\epsilon }=C^{n_{\epsilon }}$ of the epsilon sector, where we assume that a finite number $n_{\epsilon }$ of lattices $\Gamma_{\epsilon ,\lambda }$—introduced in (Equation 26)—are allowed. Concretely, and for simplicity, we will take $n_{\epsilon }=2$ with two alternative values $\epsilon_{\pm }$. With such simple choices, we find that ${ℋ}_{\epsilon }={ℋ}_{in}={ℋ}_{out}=C^{2}$, i.e., with the exception of ${ℋ}_{m}$ that is infinite dimensional, the other quantum numbers have been truncated to standard *q*-bits. Thus, a basis of this Hilbert space is defined by the set of vectors |m,ε,i,o〉 with $m\in R^{+}$, ε (encoding the microscopic ϵ sectors), *i* (encoding the *inside* particle state) and *o* (encoding the *outside* state) spin-like quantum numbers taking values ±. The inner product between basis states is
(39)$$\langle m_{1},\varepsilon_{1},i_{1},o_{1}|m_{2},\varepsilon_{2},i_{2},o_{2}\rangle =\delta \left(m_{1}-m_{2}\right)\delta_{\varepsilon_{1},\varepsilon_{2}}\delta_{i_{1},i_{2}}\delta_{o_{1},o_{2}}\;.$$
After having defined the truncation of the Hilbert space, we introduce the initial state representing a toy version of the in-vacuum in the context of Hawking radiation, namely
(40)$$|\psi_{0}\rangle =\sum_{\begin{array}{c} i=\pm \\ \varepsilon =\pm \end{array}}\int dm\;\psi_{0}\left(m,a_{0}+\epsilon_{\varepsilon }\right)|m,\varepsilon ,i,i\rangle ,$$
and the normalization condition $\langle \psi_{0}|\psi_{0}\rangle =1$, implying
(41)$$\int dm\;|\psi_{0}\left(m,a_{0}+\epsilon_{\varepsilon }\right){|}^{2}=\frac{1}{4}\;.$$
The key feature of the initial state is that it represents a state in which the excitations in ${ℋ}_{in}$ and ${ℋ}_{out}$ are maximally correlated as in the standard in-vacuum. We can then apply the evolution law (Equation 29) to this state, a shift by nλ is given by
(42)$$\begin{array}{cc} |\psi \rangle & =\sum_{\begin{array}{c} i=\pm \\ \varepsilon =\pm \end{array}}\int dm\;e^{\frac{i}{2}\log\left(1+\frac{n\lambda }{a_{0}+\epsilon_{\varepsilon }}\right)\left(m+\frac{p_{i}^{2}}{16\pi \ell_{p}^{2}m}\right)}\psi_{0}\left(m,a_{0}+\epsilon_{\varepsilon }\right)|m,\epsilon_{\varepsilon },i,i\rangle \end{array}$$
(43)$$\begin{array}{cc} \; & =\sum_{\begin{array}{c} i=\pm \\ \varepsilon =\pm \end{array}}\int dm\;C\left(m,\epsilon_{\varepsilon },i,n\right)\;|m,\epsilon_{\varepsilon },i,i\rangle \;, \end{array}$$
where, in order to simplify the notation, we introduced
(44)$$C\left(m,\epsilon_{\varepsilon },i,n\right)=e^{\frac{i}{2}\log\left(1+\frac{n\lambda }{a_{0}+\epsilon_{\varepsilon }}\right)\left(m+\frac{p_{i}^{2}}{16\pi \ell_{p}^{2}m}\right)}\psi_{0}\left(m,a_{0}+\epsilon_{\varepsilon }\right)\;.$$
These are the quantum gravitational dynamics of the initial state (of maximally correlated Hawking partners). These dynamics encode the back reaction of the Hawking particles with the quantum geometry exactly (in the sense of our simplified model). It only remains to study the dynamical evolution of the entanglement between the relevant degrees of freedom as the *inside* partner evolves toward the strong quantum gravity regime (the singularity) inside the black hole.

### 3.2. The Subsystems of Interest and Their Reduced Density Matrices

We now have access to the density matrix of the system ρ=|ψ〉〈ψ|, which is a pure state at any area *a*-time (by the unitarity of the quantum evolution Equation 29). Its expression is given by
(45)$$\rho =\sum_{\begin{array}{c} i,j=\pm \\ \varepsilon ,\varepsilon^{\prime }=\pm \end{array}}\int dmdm^{\prime }\;C\left(m,\epsilon_{\varepsilon },i,n\right)\bar{C(m^{\prime },\epsilon_{\varepsilon^{\prime }},j,n)}|m,\epsilon_{\varepsilon },i,i\rangle \langle m^{\prime },\epsilon_{\varepsilon^{\prime }},j,j|.$$
Now, we determine the reduced density matrices of the relevant subsystems. We will start by writing the density matrix of the subsystem ${ℋ}_{\epsilon }⊗{ℋ}_{in}$ obtained from tracing out the ${ℋ}_{m}$ and ${ℋ}_{out}$. One can show that
(46)$$\rho^{\epsilon ,\mathrm{in}}=\left(\begin{array}{cccc} 1/4 & 0 & D(\epsilon_{1},\epsilon_{2},+,+,n) & 0\\ 0 & 1/4 & 0 & D(\epsilon_{1},\epsilon_{2},-,-,n)\\ \bar{D(\epsilon_{1},\epsilon_{2},+,+,n)} & 0 & 1/4 & 0\\ 0 & \bar{D(\epsilon_{1},\epsilon_{2},-,-,n)} & 0 & 1/4 \end{array}\right)\;,$$
where
(47)$$D\left(\epsilon_{\varepsilon },\epsilon_{\varepsilon^{\prime }},i,j;n\right)\equiv \int dm\;C\left(m,\epsilon_{\varepsilon },i,n\right)\bar{C(m,\epsilon_{\varepsilon^{\prime }},j,n)},$$
Tracing further over ${ℋ}_{in}$, we obtain the reduced density matrix of the subsystem ${ℋ}_{\epsilon }$
(48)$$\rho^{\epsilon }=\left(\begin{array}{cc} 1/2 & D\left(\epsilon_{1},\epsilon_{2},+,+,n\right)+D\left(\epsilon_{1},\epsilon_{2},-,-,n\right)\\ \bar{D(\epsilon_{1},\epsilon_{2},+,+,n)}+\bar{D(\epsilon_{1},\epsilon_{2},-,-,n)} & 1/2 \end{array}\right)\;,$$
while tracing over ${ℋ}_{\epsilon }$, we obtain the reduced density matrix of the subsystem ${ℋ}_{in}$
(49)$$\rho^{\mathrm{in}}=\left(\begin{array}{cc} 1/2 & 0\\ 0 & 1/2 \end{array}\right)\;.$$
One can notice that this last density matrix does not depend on time, which means that the entanglement entropy of this subsystem is constant. More generally, the results (Equation 46), (Equation 48) and (Equation 49) will be used in the next section to study the correlations between the two subsystems ${ℋ}_{\epsilon }$ and ${ℋ}_{in}$. But we are also interested in the evolution of the correlations between the subsystems ${ℋ}_{in}$ and ${ℋ}_{out}$. Consequently, we need the reduced density matrix associated to the subsystem ${ℋ}_{in}⊗{ℋ}_{out}$. It follows that
(50)$$\rho^{\mathrm{in},\mathrm{out}}=\left(\begin{array}{cccc} 1/2 & 0 & 0 & D\left(\epsilon_{1},\epsilon_{1},-,+,n\right)+D\left(\epsilon_{2},\epsilon_{2},-,+,n\right)\\ 0 & 0 & 0 & 0\\ 0 & 0 & 0 & 0\\ \bar{D\left(\epsilon_{1},\epsilon_{1},-,+,n\right)+D\left(\epsilon_{2},\epsilon_{2},-,+,n\right)} & 0 & 0 & 1/2 \end{array}\right)\;.$$
Finally, the last ingredient that one will need is the matrix density of the subsystem ${ℋ}_{out}$ given by
(51)$$\rho^{\mathrm{out}}=\left(\begin{array}{cc} 1/2 & 0\\ 0 & 1/2 \end{array}\right)\;.$$

### 3.3. Computation of the Interference Terms

Based on the previous section, we have a complete knowledge for any *a* of the total system, cf (Equation 45), and also of the relevant subsystems ${ℋ}_{in}$ (matter inside), ${ℋ}_{out}$ (matter outside), ${ℋ}_{\epsilon }$ (quantum hair), ${ℋ}_{in}⊗{ℋ}_{\epsilon }$, and ${ℋ}_{in}⊗{ℋ}_{out}$, respectively, given by (Equation 49), (Equation 51), (Equation 48), (Equation 46) and (Equation 50). We now want to study the entanglement of these subsystems with their environment, which is encoded in the non-diagonal terms of the different density matrices characterized by the complex function $D(\epsilon ,\epsilon^{\prime },i,j;n)$ appearing in (Equation 46), (Equation 48) and (Equation 50). In order to evaluate $D(\epsilon ,\epsilon^{\prime },i,j;n)$, we need to choose the form of initial wave function $\psi_{0}\left(m,a_{0},p_{i}\right)$ of the quantum geometry in (Equation 40). We take the following semiclassical state
(52)$$\psi_{0}\left(m,a_{0}+\epsilon_{\varepsilon }\right)=\frac{1}{2\sqrt{\sigma \sqrt{\pi }}}e^{-\frac{{\left(m-m_{0}\right)}^{2}}{2\sigma^{2}}},$$
normalized according to (Equation 41), and centered at $m=m_{0}$ (with a spread 2σ) and $p_{m}=0$. Moreover, we assume that $m_{0}\gg \sigma$. According to (Equation 34)—giving initial conditions at $a=100\lambda \ell_{p}^{2}$ (an arbitrary choice of initial time)—such state corresponds to a semiclassical state of the spacetime geometry corresponding to a Schwarzschild black hole of mass
(53)$$M=\frac{\sqrt{4\pi }\ell_{p}^{3}m_{0}^{2}}{5\ell_{0}^{2}\sqrt{\lambda }}.$$
The relevant amplitude $D(\epsilon ,\epsilon^{\prime },i,j)$ becomes
(54)$$D\left(\epsilon_{\varepsilon },\epsilon_{\varepsilon^{\prime }},i,j;n\right)=\frac{1}{4\sigma \sqrt{\pi }}\int_{-\infty }^{+\infty }dm\;e^{-\frac{{\left(m-m_{0}\right)}^{2}}{\sigma^{2}}+\frac{i}{2}\log\left(1+\frac{n\lambda }{a_{0}+\epsilon_{\varepsilon }}\right)\left(m+\frac{p_{i}^{2}}{16\pi \ell_{p}^{2}m}\right)-\frac{i}{2}\log\left(1+\frac{n\lambda }{a_{0}+\epsilon_{\varepsilon^{\prime }}}\right)\left(m+\frac{p_{j}^{2}}{16\pi \ell_{p}^{2}m}\right)}.$$
In order to compute the previous integral, we first change variables to $k=m-m_{0}$, with k∈[−∞,+∞] and approximate the integral using the stationary phase method (Gaussian integrals). The result is
(55)$$\begin{array}{cc} D(\epsilon_{\varepsilon },\epsilon_{\varepsilon^{\prime }},i,j;n)\approx & \frac{e^{\frac{1}{2}i\left(\frac{p_{i}^{2}}{16\pi \ell_{p}^{2}m_{0}}+m_{0}\right)\log\left(\frac{a_{0}+\lambda n+\epsilon_{\varepsilon }}{a_{0}+\epsilon_{\varepsilon }}\right)-\frac{1}{2}i\left(\frac{p_{j}^{2}}{16\pi \ell_{p}^{2}m_{0}}+m_{0}\right)\log\left(1+\frac{n\lambda }{a_{0}+\epsilon_{\varepsilon^{\prime }}}\right)}}{4\sigma \sqrt{\pi }}\\ \; & \times \int_{-\infty }^{+\infty }e^{k^{2}\left(\frac{ip_{i}^{2}\log\left(1+\frac{n\lambda }{a_{0}+\epsilon_{\varepsilon }}\right)}{32\pi \ell_{p}^{2}m_{0}^{3}}-\frac{ip_{j}^{2}\log\left(1+\frac{n\lambda }{a_{0}+\epsilon_{\varepsilon^{\prime }}}\right)}{32\pi \ell_{p}^{2}m_{0}^{3}}-\frac{1}{\sigma^{2}}\right)}\\ \; & \times e^{k\left(\frac{1}{2}i\left(1-\frac{p_{i}^{2}}{16\pi \ell_{p}^{2}m_{0}^{2}}\right)\log\left(1+\frac{n\lambda }{a_{0}+\epsilon_{\varepsilon }}\right)-\frac{1}{2}i\left(1-\frac{p_{j}^{2}}{16\pi \ell_{p}^{2}m_{0}^{2}}\right)\log\left(\frac{a_{0}+\lambda n+\epsilon_{\varepsilon^{\prime }}}{a_{0}+\epsilon_{\varepsilon^{\prime }}}\right)\right)}dk. \end{array}$$

This integral is analytically computable and is equal to
(56)$$\begin{array}{cc} D(\epsilon_{\varepsilon },\epsilon_{\varepsilon^{\prime }},i,j;n)\approx & \frac{1}{\sigma \sqrt{-\frac{ip_{i}^{2}\log\left(1+\frac{n\lambda }{a_{0}+\epsilon_{\varepsilon }}\right)}{\ell_{p}^{2}m_{0}^{3}}+\frac{ip_{j}^{2}\log\left(1+\frac{n\lambda }{a_{0}+\epsilon_{\varepsilon^{\prime }}}\right)}{\ell_{p}^{2}m_{0}^{3}}+\frac{32\pi }{\sigma^{2}}}}\\ \; & \times \exp\left(-\frac{\sigma^{2}{\left(\left(p_{i}^{2}-16\pi \ell_{p}^{2}m_{0}^{2}\right)\log\left(1+\frac{n\lambda }{a_{0}+\epsilon_{\varepsilon }}\right)-\left(p_{j}^{2}-16\pi \ell_{p}^{2}m_{0}^{2}\right)\log\left(1+\frac{n\lambda }{a_{0}+\epsilon_{\varepsilon^{\prime }}}\right)\right)}^{2}}{128\pi \ell_{p}^{2}m_{0}\left(-ip_{i}^{2}\sigma^{2}\log\left(1+\frac{n\lambda }{a_{0}+\epsilon_{\varepsilon }}\right)+ip_{j}^{2}\sigma^{2}\log\left(1+\frac{n\lambda }{a_{0}+\epsilon_{\varepsilon^{\prime }}}\right)+32\pi \ell_{p}^{2}m_{0}^{3}\right)}\right)\\ \; & \times \sqrt{2\pi }{\left(1+\frac{\lambda n}{a_{0}+\epsilon_{\varepsilon }}\right)}^{\frac{1}{2}i\left(\frac{p_{i}^{2}}{16\pi \ell_{p}^{2}m_{0}}+m_{0}\right)}{\left(1+\frac{\lambda n}{a_{0}+\epsilon_{\varepsilon^{\prime }}}\right)}^{-\frac{1}{2}i\left(\frac{p_{j}^{2}}{16\pi \ell_{p}^{2}m_{0}}+m_{0}\right)}. \end{array}$$

## 4. Evolution of the Entanglement and Correlations between the Different Subsystems

We have all we need to analyze the evolution of entanglement between the relevant degrees of freedom as the system evolves toward the singularity at a=0. The question that remains is what measure of entanglement we use for interpretation. Here, there are several alternatives which all encode the same physics in different terms. The simplest (yet to some extent basis-dependent) method is the use of the so-called decoherence function. Mutual information is basis independent and provides clearer ground for interpretation and will be defined in Section 4.2.

### 4.1. Decoherence Function

It is often said that the entanglement of a system with its environment tends to kill the off diagonal terms of its density matrix. The consequence is that a system that is initially pure becomes mixed due to this interaction. It is possible to make a more precise statement thanks to the decoherence functions. Let us consider a general subsystem ρ(t) written in a basis that we will note ${\left\{|n\rangle \right\}}_{n\in N}$. This subsystem is initially pure and interacts with its environment. The decoherence functions will be noted $\Gamma_{nm}\left(t\right)$ and are defined as follows.
(57)$$|\langle n|\rho \left(t\right)|m\rangle |=\exp(\Gamma_{nm}\left(t\right)).$$
These functions describe the time evolution of the off-diagonal terms of the density matrix and therefore encode the evolution of a subsystem from a pure state to a mixed one. For a system of dimension N there are N(N−1)/2 independent off diagonal terms in the density matrix, and as many decoherence functions. One can notice that these functions depend on the basis we chose to write the density matrix. There is, however a natural basis, the ‘energy basis’, in which the diagonal terms are time independent. We will work in this basis here. We are interested first in the evolution of the purity of the ϵ subsystem (i.e., the Planckian geometrical degrees of freedom). One can see that it is easy to obtain the decoherence function for this subsystem, which is unique, from the expression (Equation 48)
(58)$$\Gamma \left(n\lambda \right)=|D\left(\epsilon_{1},\epsilon_{1},-,+,n\right)+D\left(\epsilon_{2},\epsilon_{2},-,+,n\right)|.$$
One can now plot the evolution of this function in terms of the time variable nλ in order to illustrate how the purity of the subsystem ϵ evolves, and how this evolution depends on the different parameters of the system $m_{0},p_{+},\sigma$ (cf Figure 3).

### 4.2. Mutual Information as an Entanglement Measure

The previous was a quick illustration about how the microscopic states actually entangle during the dynamical evolution and hence cannot be neglected in discussions of unitarity. However, the method used is not adapted for a clear interpretation due to its intrinsic basis-dependent nature. Here, we interpret the dynamical evolution of entanglement using a basis-independent measure of entanglement: mutual information. We first briefly remind the reader of a few definitions and then apply them to our system. The first ingredient we need is the notion of relative entropy. Relative entropy is a entanglement measure defined between two states described by the density matrix ρ and σ in the following way
(59)S(ρ|σ):=Tr(ρlogρ−ρlogσ).
One can show that S(ρ|σ)≥0 and it is equal to zero when ρ=σ. It quantifies the distinguishability of ρ from σ. An important property of the relative entropy is that for any bounded operator $\hat{O}$, the relative entropy satisfies
(60)$$\frac{{\left(\mathrm{Tr}\left(\hat{O}\rho \right)-\mathrm{Tr}\left(\hat{O}\sigma \right)\right)}^{2}}{\left|\right|\hat{O}\left|\right|}\leq S\left(\rho |\sigma \right),$$
with $\left|\right|\hat{O}\left|\right|=\sup\sqrt{\langle \psi |\hat{O}|\psi \rangle /\langle \psi ||\psi \rangle }$. The mutual information is a specific case of the relative entropy. Let us consider a system which can be decomposed into three subsystems: *A*, *B* and *C*. We assume that the total system is in a pure state |ψ〉, the subsystem A+B is described by the density matrix $\rho_{AB}$ and the two subsystem *A* and *B* are, respectively, described by the density matrix $\rho_{A}$ and $\rho_{B}$. The mutual information (between *A* and *B* when *C* is traced out) is given by
(61)$$I_{AB|C}:=S\left(\rho_{AB}|\rho_{A}⊗\rho_{B}\right).$$
The mutual information is therefore also an entanglement measure and quantifies how much a mixed state is distinguishable from the uncorrelated state that we can obtain by treating the different subsystems as independent. By using (59), one can show that
(62)$$I_{AB|C}=S_{A}+S_{B}-S_{AB},$$
where *S* is the usual Von Neumann entropy. An important property of the mutual information comes from (60), which, in the case $\rho =\rho_{AB}$ and $\sigma =\rho_{A}⊗\rho_{B}$ gives, for all $\hat{O}={\hat{O}}_{A}⊗{\hat{O}}_{B}$ such that ${\hat{O}}_{A}$ and ${\hat{O}}_{B}$ are bounded, the following relation
(63)$$\frac{{\left(\langle \psi |{\hat{O}}_{A}{\hat{O}}_{B}|\psi \rangle -\langle \psi |{\hat{O}}_{A}|\psi \rangle \langle \psi |{\hat{O}}_{B}|\psi \rangle \right)}^{2}}{2||{\hat{O}}_{A}\left|||\right|{\hat{O}}_{B}\left|\right|}\leq I_{AB|C}\left(|\psi \rangle \right).$$
This relation means that the mutual information is an upper bound of the correlations between the subsystems *A* and *B*. Thus, we will use this entanglement measure to track the correlations between the different subsystems in our model because it has a clear physical meaning in terms of entanglement and correlations.

#### 4.2.1. Evolution of the Correlations between the Hawking Partners

We can compute the mutual information between the inside and the outside Hawking excitations in the matter sector
(64)$$I_{\mathrm{in},\mathrm{out}}(|\psi \rangle )=S\left(\rho^{\mathrm{in}}\right)+S\left(\rho^{\mathrm{out}}\right)-S\left(\rho^{\mathrm{in},\mathrm{out}}\right)$$
From (Equation 49), (Equation 51) and (Equation 50), it follows that
(65)$$\begin{array}{cc} I_{\mathrm{in},\mathrm{out}}(|\psi \rangle ) & =\frac{1}{2}\left(1-2|D\left(\epsilon_{1},\epsilon_{1},j,0,n\right)+D\left(\epsilon_{2},\epsilon_{2},j,0,n\right)|\right)\log\left(\frac{1}{2}\left(1-2|D\left(\epsilon_{1},\epsilon_{1},j,0,n\right)+D\left(\epsilon_{2},\epsilon_{2},j,0,n\right)|\right)\right)\\ \; & +\frac{1}{2}\left(1+2|D\left(\epsilon_{1},\epsilon_{1},j,0,n\right)+D\left(\epsilon_{2},\epsilon_{2},j,0,n\right)|\right)\log\left(\frac{1}{2}\left(1+2|D\left(\epsilon_{1},\epsilon_{1},j,0,n\right)+D\left(\epsilon_{2},\epsilon_{2},j,0,n\right)|\right)\right)\\ \; & +\log(4), \end{array}$$
which is the final explicit result if we use (Equation 56). We can now look at the evolution of the mutual information between the inside and outside Hawking pairs as a function of $a=n\lambda \ell_{p}^{2}$, for $n\lambda \ell_{p}^{2}\in \left[0,a_{0}-2\lambda \ell_{p}^{2}\right]$. In other words, we can visualize the evolution of the entanglement/correlations between these two subsystems when the area decreases. As we can anticipate, this quantity will depend on the choice of $m_{0}$ (recall that the mass $M\propto m_{0}^{2}$ from (Equation 53)), on the spread of the wave packet σ, and on the square of the ‘characteristic energy’ of the Hawking particle *j*.

The results are shown in Figure 4, where we observe the decrease in the mutual information between the Hawking partners (initially maximally correlated as in Hawking radiation) as the inside partner evolves toward the singularity. This is an expected feature: due to the back reaction of the inside particle on the quantum geometry, correlations are transferred to degrees of freedom other than those encoded in ${ℋ}_{in}$. This tells us that an appropriate account of unitarity necessitates the inclusion of all the degrees of freedom. From such a general perspective, the result is expected; a trivial implication of the monogamy of entanglement in quantum mechanics. The non-trivial feature here, which will become clear in Section 4.2.2, is that essential correlations are established with the microscopic degrees of freedom (modeled here by the ϵ sectors).

Figure 4 shows that the decoherence effect (the decrease in the mutual information as the inside partner approaches the singularity) is ‘faster’ when the black hole is smaller (when $m_{0}$ decreases). The effect is less important initially for macroscopic black holes, even when it always becomes relevant when the singularity approaches. Larger $p_{\phi }$ increases the effect, as expected from the fact that this increases the back reaction on the geometry and microscopic sectors. The dependence on the spread σ is consistent with the present analysis.

#### 4.2.2. Evolution of the Correlations between the in Falling Matter and the ϵ d.o.f

We can then compute the mutual information between the ingoing and the ϵ degrees of freedom
(66)$$I_{\mathrm{in},\epsilon }(|\psi \rangle )=S\left(\rho^{\mathrm{in}}\right)+S\left(\rho^{\epsilon }\right)-S\left(\rho^{\mathrm{in},\epsilon }\right)$$
As in the previous subsection, it follows
(67)$$\begin{array}{cc} I_{\mathrm{in},\epsilon }(|\psi \rangle ) & =\frac{1}{4}(1-4|D\left(\epsilon_{1},\epsilon_{2},0,0,n\right)\left|\right)\log\left(\frac{1}{4}-\left|D\left(\epsilon_{1},\epsilon_{2},0,0,n\right)\right|\right)\\ \; & +\frac{1}{4}(1+4|D\left(\epsilon_{1},\epsilon_{2},0,0,n\right)\left|\right)\log\left(\frac{1}{4}+\left|D\left(\epsilon_{1},\epsilon_{2},0,0,n\right)\right|\right)\\ \; & +\frac{1}{4}(1-4|D\left(\epsilon_{1},\epsilon_{2},j,j,n\right)\left|\right)\log\left(\frac{1}{4}-\left|D\left(\epsilon_{1},\epsilon_{2},j,j,n\right)\right|\right)\\ \; & +\frac{1}{4}(1+4|D\left(\epsilon_{1},\epsilon_{2},j,j,n\right)\left|\right)\log\left(\frac{1}{4}+\left|D\left(\epsilon_{1},\epsilon_{2},j,j,n\right)\right|\right)\\ \; & +\frac{1}{2}(2|D\left(\epsilon_{1},\epsilon_{2},0,0,n\right)+D\left(\epsilon_{1},\epsilon_{2},j,j,n\right)\left|-1\right)\log\left(\frac{1}{2}-\left|D\left(\epsilon_{1},\epsilon_{2},0,0,n\right)+D\left(\epsilon_{1},\epsilon_{2},j,j,n\right)\right|\right)\\ \; & -\frac{1}{2}(2|D\left(\epsilon_{1},\epsilon_{2},0,0,n\right)+D\left(\epsilon_{1},\epsilon_{2},j,j,n\right)\left|+1\right)\log\left(\frac{1}{2}+\left|D\left(\epsilon_{1},\epsilon_{2},0,0,n\right)+D\left(\epsilon_{1},\epsilon_{2},j,j,n\right)\right|\right)\\ \; & +\log(2)\;, \end{array}$$
which can be evaluated using (Equation 56). The results are reported in Figure 5. We see that, as the correlations between the outside and inside Hawking excitations decrease (Figure 4), the correlations of the inside partner and the microscopic degrees of freedom increase as it evolves toward the strong quantum gravity regime close to the singularity. As in the previous subsection, the dependence on the various parameters is the expected one.

#### 4.2.3. Evolution of the Correlations between the Out Going and the ϵ d.o.f

In the previous subsection, we studied the correlations between the inside Hawking partner and the microscopic ϵ sector (the one representing Planckian granularity in our toy model). Correlations grow as the partner falls into the singularity as its back reaction on the microscopic structure of the quantum geometry becomes more and more important as the singularity approaches. A quantum counterpart of this is the fact that the outside partner escaping at infinity will correlate with the quantum degrees of freedom of the geometry as well, even if it never comes into direct interaction with them (this is a consequence of the monogamy of entanglement). Indeed it is very easy to relate (in our system) the mutual information $I_{\mathrm{in},\epsilon }(|\psi \rangle )$ with $I_{\mathrm{out},\epsilon }(|\psi \rangle )$. To see this, one can first start from the definition of the mutual information between the subsystem corresponding to the outside partner and ϵ, namely
(68)$$I_{\mathrm{out},\epsilon }(|\psi \rangle )=S\left(\rho^{\mathrm{out}}\right)+S\left(\rho^{\epsilon }\right)-S\left(\rho^{\mathrm{out},\epsilon }\right).$$
By comparing the density matrices (Equation 49) and (Equation 51), one obtains that
(69)$$I_{\mathrm{in},\epsilon }(|\psi \rangle )=I_{\mathrm{out},\epsilon }(|\psi \rangle ).$$
Therefore, correlations between outside partner and the microscopic ϵ sector appear at the same rate as those between the inside partner and the ϵ sector.

#### 4.2.4. Interpretation

The results in this section consistently show how the information coded in the initial correlations between the Hawking partners in ${ℋ}_{in}⊗{ℋ}_{out}$ are transferred to the other degrees of freedom involved. This is expected to be the case due to the gravitational interaction between the matter sector and the spacetime geometry. What is particularly interesting, and novel in our case, is the generic transfer of entanglement to the ‘quantum hair’ represented in our simple model by ${ℋ}_{\epsilon }$. The computations show clearly, in the results reported in Figure 3, Figure 4 and Figure 5, that correlations are established with these degrees of freedom during the evolution of the inside excitation toward the singularity. This indicates that, in a process where the mass of the black hole would be completely disappear via Hawking evaporation (a process that our simple model cannot account for), unitarity could be maintained via the purification of Hawking radiation granted by degrees of freedom, like the ϵ sector of this toy model. Despite the simplistic nature of the model, we notice that it complies with the expected behaviour of the back reaction: the lower the mass (the smaller $m_{0}$), or equivalently the smaller the black hole, the stronger the development of entanglement between the matter and the defects in the quantum geometry (stronger interaction with the Planckian granularity). The effect also grows with the particle momentum $p_{\phi }$ and dispersion σ, as expected.

## 5. Discussion

The problem of unitarity in black hole formation and evaporation necessitates a deep understanding of the strong quantum gravity regime near the singularity beyond the black hole horizon. This presents a challenge for both the *holographic* and *non-holographic* perspectives. In the holographic view, macroscopic black holes are believed to possess a finite number of internal degrees of freedom bounded by their horizon area. Consequently, they are expected to follow a standard Page curve, wherein the exterior entropy starts decreasing at the macroscopic stage known as the Page time. Conversely, in the non-holographic perspective, which is the framework of this work, the number of internal degrees of freedom inside black holes is unrelated to their horizon area. In this scenario, the exterior entropy continues to grow while the black hole is macroscopic, and the purification occurs after the semiclassical regime. In non-holographic scenarios, where information is recovered after the complete evaporation of a black hole, it becomes essential to identify the appropriate purifying degrees of freedom. Subsequently, understanding how the correlations between the Hawking partners are dynamically transferred to these degrees of freedom becomes crucial.

Suitable degrees of freedom are those that can persist after complete evaporation without significantly affecting the energy budget, considering that only a Planck mass energy is available for the multitude of degrees of freedom needed to purify the emitted Hawking radiation until the black hole’s total evaporation. Approaches to quantum gravity, which postulate a discrete fundamental nature of matter and geometry at the Planck scale, offer a natural set of such degrees of freedom. In these theories, the concepts of smooth geometry and continuous fields are approximate and emerge only when probed with sufficiently coarse measurements. Within this framework, flat configurations such as Minkowski spacetime with the Minkowski vacuum for matter fields are expected to exhibit high degeneracy in a coarse-grained sense, meaning that a single macroscopic configuration corresponds to a large number of microscopic states. Any deviations among these microscopic states can be viewed as defects in the fundamental structure that are indistinguishable at the macroscopic level. These precisely are the types of degrees of freedom suitable for purification. They are numerous (due to their local nature and increasing number with spatial extension) and, in the aforementioned sense, possess negligible weight or influence on the system’s overall energy.

Once suitable degrees of freedom have been identified, one needs to understand the dynamical process by which unitarity is preserved when these are tracked during the black hole formation and evaporation. In particular, while the black hole is macroscopic and emits Hawking radiation, one needs to understand the mechanism by which the (initially) maximally correlated Hawking pairs evolve into a particle, escaping to infinity as Hawking radiation correlated to the Planckian degrees of freedom inside that will emerge (after complete evaporation) as the purifying defects of the previous paragraph.

Both ingredients are shown to be present in the simple model of quantum black hole analyzed in this work. The model is expected to capture the dynamics of an infalling scalar excitation when it is spherically symmetric in the near singularity approximation. It can be used to model the near-singularity dynamics of a Hawking pair of scalar particles, including the back reaction of the inside partner as it approaches the near-singularity regime. On the one hand, the toy model used here mimics the quantization of gravity in the framework of loop quantum gravity and thus contains the type of microscopic ‘weightless’ degrees of freedom. More precisely, the spectrum of the mass operator is infinitely degenerate, due to the existence of quantum hair associated with what we call the ϵ sectors, Section 2.5. On the other hand, we showed explicitly how correlations are transferred dynamically to the ‘hidden’ quantum hair as the inside excitation approaches the near singularity-regime. This was carried out first by using decoherence function (suitable intuitive method for the simplified setup of the toy model because there is a single such function), and later using the basis-independent mutual information entanglement measure.

Of course, the model does not explain how the black hole evaporates and how the information remains after complete evaporation coded in the fundamental ‘hair’. This would possibly require the extremely non-trivial step of allowing local degrees of freedom and treating their strong quantum dynamics near the singularity. We are not sure if one can find intermediate models between our simplistic one and the fully general situation that requires treatment without any possible approximation. Yet, even when one is still far from the full understanding of the process of black hole formation and evaporation (in approaches like loop quantum gravity and others), we hope that the indications provided by models, like the one studied here, stimulate possibilities and new ideas that could prove useful for researchers investigating this central problem.

## Figures and Tables

**Figure 1 entropy-25-01479-f001:**
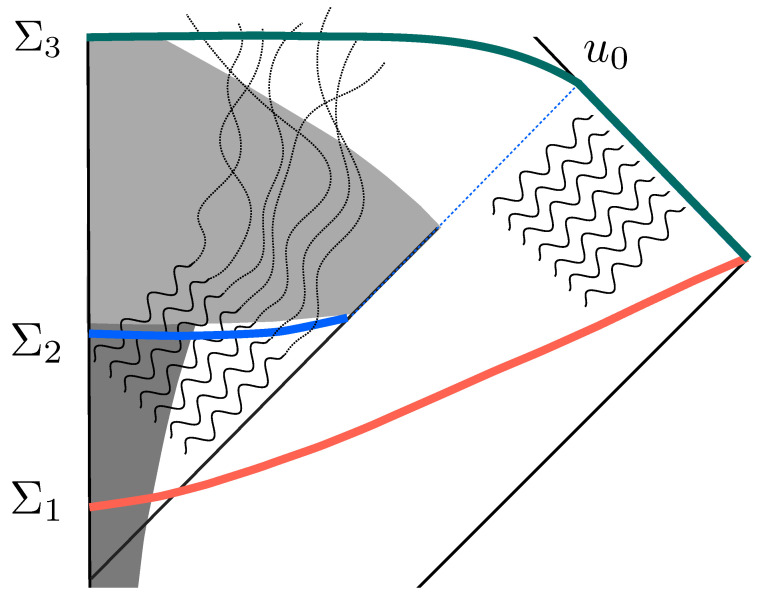
Inside Hawking partners (initially maximally correlated with the Hawking radiation emitted to ${ℐ}^{+}$) fall into the quantum gravity region where they are dynamically forced to interact with the microscopic granularity of quantum gravity. Near the singularity, matter excitations are infinitely blue shifted into the Planckian regime. Such interactions transfer the entanglement from the initial field theoretic degrees of freedom in the Hawking partner to defects in the microscopic structure at the Planck scale. Defects emerge in the future of the quantum gravity region and their entanglement with the Hawking radiation purifies the final state. As the defects do not weight (the mean field flat geometry is degenerate), the process is compatible with energy conservation.

**Figure 2 entropy-25-01479-f002:**
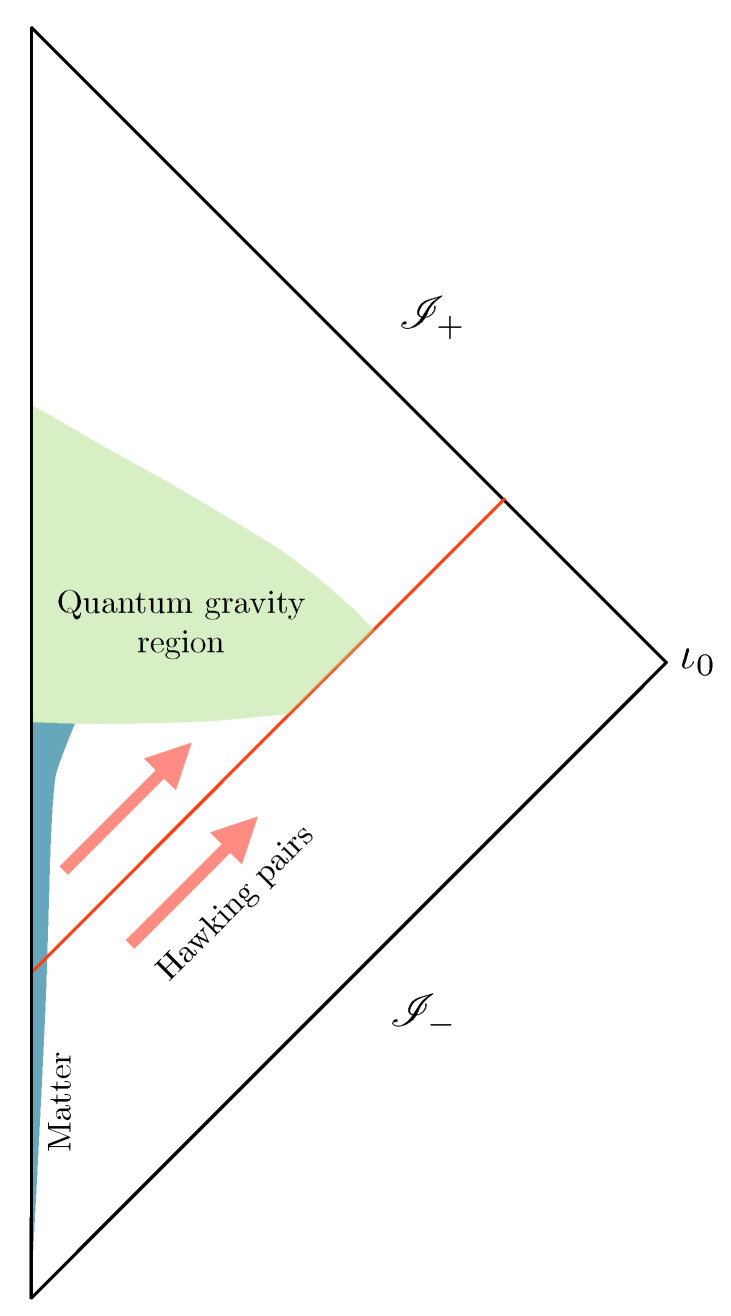
The Ashtekar–Bojowald paradigm: the inside singularity is replaced by a quantum gravity region where no semiclassical or spacetime description is possible. Geometric observables (as well as matter degrees of freedom) cannot not be represented in terms of smooth field theoretic notions. Evolution across this region is expected to be well defined in the fundamental theory into a future where geometry and matter fields are well described (in a mean field sense) by a nearly flat spacetime geometry in a vacuum.

**Figure 3 entropy-25-01479-f003:**
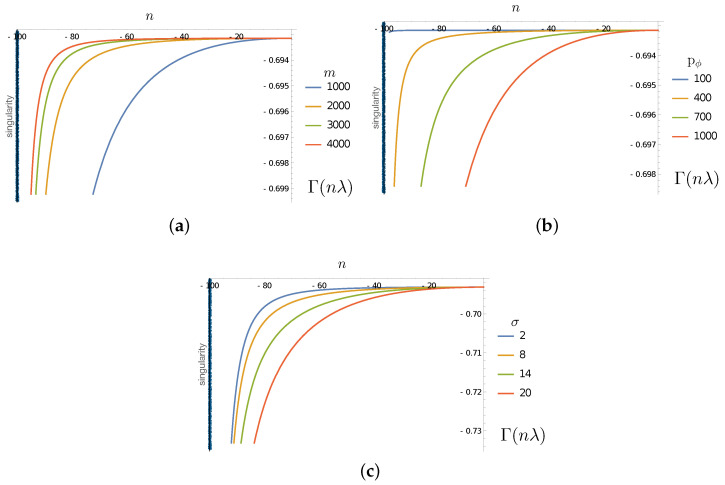
(**a**) Plot of Γ in terms of *n* evolution steps for σ=10, $p_{+}=1000$ and for different values of $m_{0}$. (**b**) Plot of Γ in terms of *n* evolution steps for $m_{0}=1000$, σ=10 and for different values of $p_{+}$. (**c**) Plot of Γ for $m_{0}=1000$, $p_{+}=1000$ and for different values of σ. The singularity corresponds to the value nλ=−100 and is indicated by a black vertical line.

**Figure 4 entropy-25-01479-f004:**
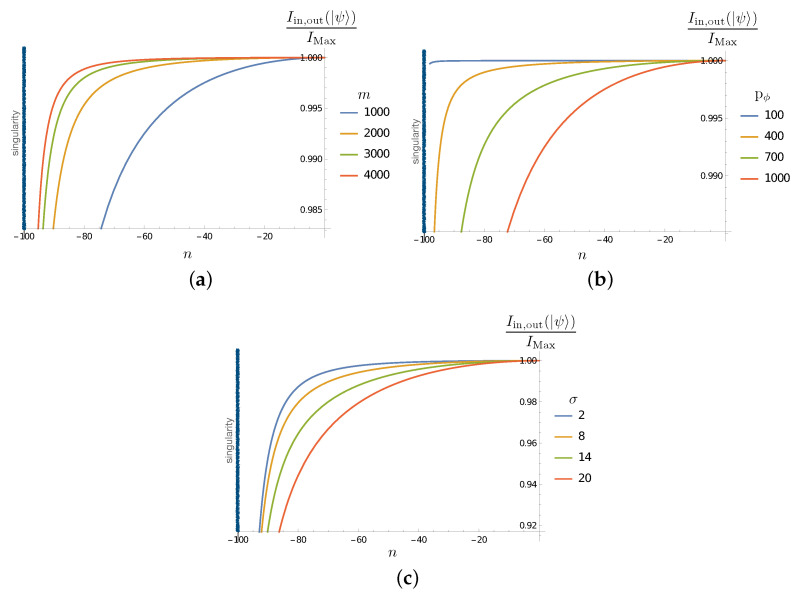
(**a**) Plot of $I_{\mathrm{in},\mathrm{out}}(|\psi \rangle )/I_{\mathrm{Max}}$ in term of *n* evolution steps for σ=10, $p_{+}=1000$ and for different values of $m_{0}$. (**b**) Plot of $I_{\mathrm{in},\mathrm{out}}(|\psi \rangle )/I_{\mathrm{Max}}$ in terms of *n* evolution steps for $m_{0}=1000$, σ=10 and for different values of $p_{+}$. (**c**) Plot of $I_{\mathrm{in},\mathrm{out}}(|\psi \rangle )/I_{\mathrm{Max}}$ in terms of *n* evolution steps for $m_{0}=1000$, $p_{+}=1000$ and for different values of σ. The singularity corresponds to the value nλ=−100 and is indicated by a black vertical line.

**Figure 5 entropy-25-01479-f005:**
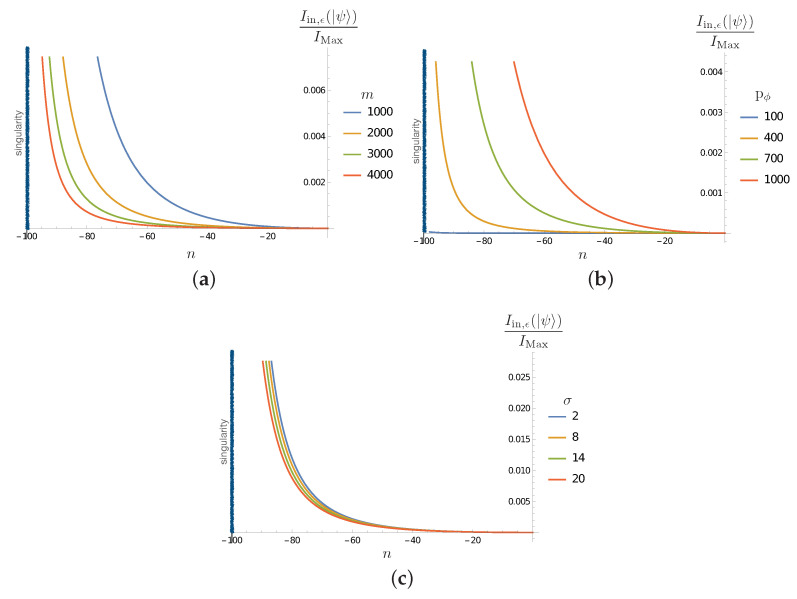
(**a**) Plot of $I_{\mathrm{in},\epsilon }(|\psi \rangle )/I_{\mathrm{Max}}$ in terms of *n* evolution steps for σ=10, $p_{+}=1000$ and for different values of $m_{0}$. (**b**) Plot of $I_{\mathrm{in},\epsilon }(|\psi \rangle )/I_{\mathrm{Max}}$ in terms of *n* evolution steps for $m_{0}=1000$, σ=10 and for different values of $p_{+}$. (**c**) Plot of $I_{\mathrm{in},\epsilon }(|\psi \rangle )/I_{\mathrm{Max}}$ in terms of *n* evolution steps for $m_{0}=1000$, $p_{+}=1000$ and for different values of σ. The singularity corresponds to the value nλ=−100 and is indicated by a black vertical line.

## Data Availability

All data that support the findings of this study are included within the article.

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
