# Peer review of "Discreteness Unravels the Black Hole Information Puzzle: Insights from a Quantum Gravity Toy Model"

_entropy, 2023, doi:10.3390/e25111479_

Round 1

Reviewer 1 Report

The presence of microscopic degrees of freedom at the Planck scale provides a natural mechanism for resolving the black hole information puzzle. In this paper, the authors aimed to understand the mechanism by which the initially maximally correlated Hawking pairs evolve into a particle escaping to infinity as Hawking radiation correlated to the Planckian degrees of freedom inside. For this, the authors used the simple quantum black hole model inspired by loop quantum gravity to review how microscopic degrees of freedom arise in the infalling scalar quantization, and how quantum correlations between matter excitations and such microscopic degrees of freedom develop unavoidably during dynamical evolution. How correlations are transferred dynamically to the ‘hidden’ quantum hair as the inside excitation approaches the near singularity regime is shown explicitly.

The paper is well-written and cover important published literature. The findings presented in this article have significant implications for our understanding of black holes evaporation and information loss paradox. I find the manuscript suitable for publication in the Entropy.

Author Response

We thank the referee for the positive report. We have made some modifications taking into account the comments of referee 2.

Reviewer 2 Report

Please see the attached report.

Please see the attached report.

Author Response

Please see reply in the attached PDF

Reviewer 3 Report

Minor english editing is required. For example, there is a floating question mark in the text above eq. (2.34). This is not the only misspel within the manuscript and I suggest that the authors read it carefully to edit it correspondingly.

Author Response

Dear Referee, we have made several corrections including the ones you suggested. We hope the paper is suitable for publication in its present form. Thank you for your help.

Round 2

Reviewer 2 Report

PLease see attached pdf

(See attached report)

Author Response

Please read file attached

Round 3

Reviewer 2 Report

Please see attached report
